# Planning and optimising CHAT&PLAN: A conversation-based intervention to promote person-centred care for older people living with multimorbidity

Teresa K. Corbett[ID][1]*, Amanda Cummings[2], Kellyn Lee[3], Lynn Calman[4], Vicky Fenerty[5], Naomi Farrington[6], Lucy Lewis[7], Alexandra Young[8], Hilary Boddington[9], Theresa Wiseman[10], Alison Richardson[11], Claire Foster[12], Jackie Bridges[13]

1 NIHR ARC Wessex, School of Health Sciences, University of Southampton, Highfield, Southampton, United Kingdom, 2 Macmillan Survivorship Research Group, School of Health Sciences, University of Southampton, Southampton, United Kingdom, 3 School of Health Sciences, University of Southampton, Highfield, Southampton, United Kingdom, 4 Macmillan Survivorship Research Group, School of Health Sciences, University of Southampton, Southampton, United Kingdom, 5 University of Southampton Library, University of Southampton, Southampton, United Kingdom, 6 University Hospital Southampton & University of Southampton, Southampton, United Kingdom, 7 Health Education England South East, University Hospital Southampton NHS Foundation Trust and University of Southampton, Otterbourne, Winchester, United Kingdom, 8 School of Health Sciences, University of Southampton, Highfield, Southampton, United Kingdom, 9 Wessex Macmillan GP, Wessex Cancer Alliance, Southampton, United Kingdom, 10 The Royal Marsden NHS Foundation Trust and University of Southampton, Southampton, United Kingdom, 11 NIHR ARC Wessex, School of Health Sciences, University of Southampton and University Hospital Southampton NHS Foundation Trust Mailpoint, Southampton General hospital, Southampton, United Kingdom, 12 Macmillan Survivorship Research Group, School of Health Sciences, University of Southampton, Southampton, United Kingdom, 13 NIHR ARC Wessex, School of Health Sciences, University of Southampton, Southampton, United Kingdom

* t.k.corbett@soton.ac.uk, t.k.corbett2@gmail.com

**Data Availability Statement:** All relevant data are within the paper and its Supporting Information files.

## Abstract

### Background

Older people are more likely to be living with cancer and multiple long-term conditions, but their needs, preferences for treatments, health priorities and lifestyle are often not identified or well-understood. There is a need to move towards a more comprehensive person-centred approach to care that focuses on the cumulative impact of a number of conditions on daily activities and quality of life. This paper describes the intervention planning process for CHAT& PLAN™, a structured conversation intervention to promote personalised care and support self-management in older adults with complex conditions.

### Methods

A theory-, evidence- and person-based approach to intervention development was undertaken. The intervention planning and development process included reviewing relevant literature and existing guidelines, developing guiding principles, conducting a behavioural analysis and constructing a logic model. Optimisation of the intervention and its implementation involved qualitative interviews with older adults with multimorbidity (n = 8), family

**Funding:** JB received funding to support this study. Funders: National Institute of Health Research, Collaboration for Leadership in Applied Health Research and Care Wessex (NIHR CLAHRC Wessex). https://clahrc-wessex.nihr.ac.uk/ The funders had no role in study design, data collection and analysis, decision to publish, or preparation of the manuscript.

**Competing interests:** Alison Richardson is a National Institute for Health Research (NIHR) Senior Investigator. The views expressed are those of the authors and not necessarily those of the NHS, the NIHR or the Department of Health and Social Care. Naomi Farrington is funded by a National Institute for Health Research (NIHR) Clinical Lectureship (ICA-CL-2015-01-003). The views expressed are those of the authors and not necessarily those of the NHS, the NIHR or the Department of Health and Social Care. This does not alter our adherence to PLOS ONE policies on sharing data and materials.

caregivers (n = 2) and healthcare professionals (HCPs) (n = 20). Data were analysed thematically and informed changes to the intervention prototype.

## Results

Review findings reflected the importance of HCPs taking a person-centred (rather than disease-centred) approach to their work with older people living with multimorbidity. This approach involves HCPs giving health service users the opportunity to voice their priorities, then using these to underpin the treatment and care plan that follow. Findings from the planning stage indicated that taking a structured approach to interactions between HCPs and health service users would enable elicitation of individual concerns, development of a plan tailored to that individual, negotiation of roles and review of goals as individual priorities change. In the optimisation stage, older adults and HCPs commented on the idea of a structured conversation to promote person-centred care and on its feasibility in practice. The idea of a shared, person-centred approach to care was viewed positively. Concerns were raised about possible extra work for those receiving or delivering care, time and staffing, and risk of creating another "tick-box" exercise for staff. Participants concluded that anyone with the appropriate skills could potentially deliver the intervention, but training was likely to be required to ensure correct utilisation and self-efficacy to deliver to the intervention.

## Conclusions

CHAT&PLAN, a structured person-centred conversation guide appears acceptable and appealing to HCPs and older adults with multimorbidity. Further development of the CHAT&PLAN intervention should focus on ensuring that staff are adequately trained and supported to implement the intervention.

## Background

By 2040, older people will account for over two-thirds of those living with and beyond cancer (LWBC) [1]. At any age, people with cancer can experience side-effects such as pain, breathlessness, and fatigue, as well as psychological problems including anxiety, depression and loss of confidence [2]. However, older adults are at heightened risk of the side-effects of some cancer-related treatments and often lack the physiological reserves required to effectively recover from acute toxicities [3]. In addition, older people are more likely to have two or more long-term conditions (used in this paper as a definition of multimorbidity). Separately, both cancer and multimorbidity are associated with poorer health-related quality of life (HRQOL) and result in complex health and social care needs [1,4–8]. Those LWBC have a higher prevalence of multimorbidity than those without cancer [9]. Multimorbidity co-occurring with cancer is associated with reduced physical health and psychological well-being, poorer mental health, and poorer survival compared to those who have no history of cancer [8–16]. Care is also complicated in this group due the social and contextual factors associated with aging [12], including increased social isolation, frailty, and polypharmacy [1].

Older adults with multimorbidity vary in what they choose to prioritise in terms of their health, as well as in the extent of treatment burden and inconvenience they are willing to accept [13]. However, health service user needs, preferences for treatments, health priorities and lifestyle are often not identified or well-understood [1,10,16]. There is a need to move

towards a more comprehensive approach that focuses on the cumulative impact of a number of conditions on daily activities and quality of life [1,8]. Patient-centred care refers to the active engagement of health service users in shared-decision making about their healthcare [9]. The term "person-centred care" has been used to represent a more holistic approach that does not view a person as simply their symptoms and/or diagnosis [11] and focuses less on the sick-role and more on the individual who lives with a condition, aiming to help the person achieve a meaningful life [14]. Communication is a central feature of person-centred care to appreciate what the care receiver values about their life, and what their preferences and priorities are [12,14]. In addition, personalised care planning (including the core components of preparation, goal setting, action planning and review) promotes an ongoing process to identify and discuss personal needs and goals, and agree and coordinate a plan for how these goals will be met, potentially leading to better health care outcomes [17]. Health and social care support underpinned by a person-centred approach to care should facilitate self-management, enable people to cope with the experience of multiple complex conditions, help them engage with life in the community, and understand how to elicit support from local services. Person-centred care is generally considered as good practice, yet remains poorly defined and implemented [14].

This paper outlines the development of 'CHAT&PLAN[TM]', a tool designed to facilitate a person-centred conversations in health and social care. Many of the existing interventions for multimorbidity lack a theoretical framework, with uncertainties about the effectiveness of interventions for people living with multimorbidity [18]. We have therefore drawn on relevant theoretical frameworks (namely Shippee's cumulative complexity model (CCM) [19] and Burden of Treatment Theory [20]) to inform our understanding of the dynamic context of living with multimorbidity in older age, and ensure relevant factors identified in theory are addressed in the intervention developed.

The 'CHAT&PLAN' intervention was developed to minimise health-related work and maximise individuals' capacity to self-manage multimorbidity. The aim was to build an intervention based on an iterative theory-, evidence- and person-based approach [21–23]. In this paper, we seek to provide a clear description of how the intervention was planned and optimised. We describe how findings allowed 'CHAT&PLAN' to be shaped by the expectations and preferences of participants, whilst emphasising insights and methods that could be applied in other settings. Outlining the development process of novel interventions helps to minimise 'research waste' and replication of interventions unlikely to be feasible or effective [24].

## Methods

### Structure of the planning and optimisation phases

'CHAT&PLAN' was developed according to a theory-, evidence- and person-based approach to intervention development [25–27]. The information generated from this method was triangulated to inform 'guiding principles' [27] and a logic model outlining the theory underpinning the intervention. The development process involved two iterative phases: planning and optimisation (See Fig 1).

Monthly development meetings were held with a multi-disciplinary team of co-investigators including Patient and Public Involvement (PPI). Three PPI volunteers supported the planning and optimization of the intervention. PPI members' input helped to ensure methods were ethical and that participation in the study was not too burdensome for older adults and caregivers. We implemented changes based on their feedback, ensuring study and intervention materials were accessible, engaging and persuasive prior to being shared with participants. The wider management team included members with backgrounds in psychology (TC, CF, KL),

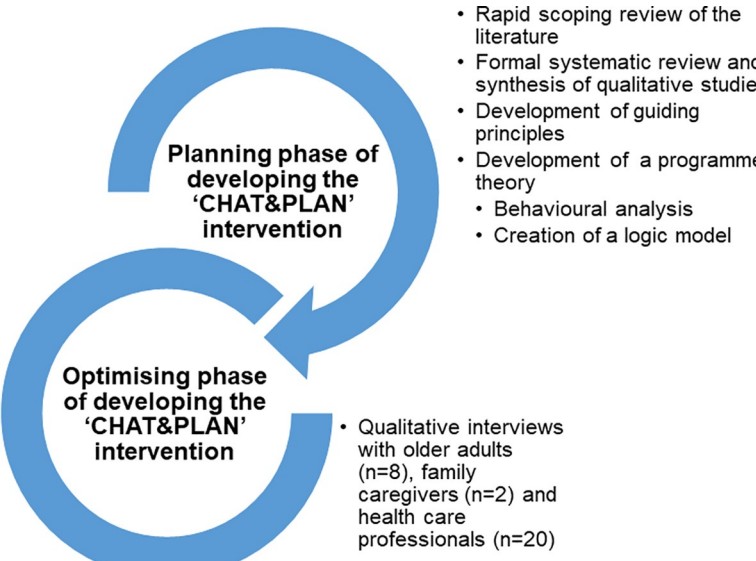

**Fig 1. Development process of the CHAT&PLAN.**

anthropology (AY), medicine (HB, AC) and nursing (LC, JW, NF, TW, AR, JB, LL), four specializing in oncology nursing (JW, NF, TW, AR) and two specializing in older adult's healthcare (JB, LL). Draft intervention materials were frequently shared for comment and iteration.

Ethical approvals were gained from the Research Integrity and Governance team, University of Southampton (ref no. 45579) and NHS London—City & East Research Ethics Committee (ref no. 253413).

## Planning phase of developing the 'CHAT&PLAN' intervention

The 'Planning' phase outlines the theory-, evidence- and person-based 'Guiding Principles' and logic model developed to underpin intervention development. This involved: an informal scoping review of relevant literature and existing guidelines, an in-depth formal qualitative literature review, development of guiding principles, and the development of a programme theory based on a behavioural analysis and a logic model.

1.  To begin, a rapid scoping review of the literature to gather evidence from a broad range of resources about potential intervention features and important contextual factors. This helped us to develop an overview of the topic area and identify key issues that were important to address. In January 2019, the following databases were used to search for studies published from 2009–2019: Medline; Web of Science; Google Scholar. Citation and snowball searching were used as well as the 'related articles' function in databases and expert recommendations. Quantitative and qualitative papers were included to explore topics of interest, including policy guidelines, reports and academic literature.

2.  Secondly, an in-depth formal systematic review and synthesis of qualitative studies was also conducted to identify what older adults living with and beyond cancer and multimorbidity report influences their self-management [28]. Databases were searched between June and July 2018 for primary qualitative research that reported older adults' perspectives on and experiences of living with cancer and multimorbidity. Further details of the methods employed are reported elsewhere [28].

3. Thirdly, guiding principles were developed to outline key intervention design objectives. These guiding principles identify user/context-specific behavioural needs and intervention features that address the design objective [27]. These were used to enhance the acceptability of an intervention and, in turn, to improve engagement and effectiveness.

4. We then developed a programme theory to define how the intervention was expected to work, by specifying the anticipated mechanisms of change involved [29]. A behavioural analysis was used to identify behaviours to be targeted by the 'CHAT&PLAN' intervention and any potential barriers and facilitators. In line with Medical Research Council (MRC) guidance [30], we constructed a logic model to illustrate the hypothesised mechanisms of action of the 'CHAT&PLAN'. This logic model was iteratively designed by the multi-disciplinary study team of co-investigators, with input from our PPI volunteers.

## Optimisation phase of developing the 'CHAT&PLAN' intervention

The 'Optimisation' phase presents qualitative findings about the experiences of older adults with other conditions alongside cancer (n = 8), family caregivers (n = 2) and health care professionals (n = 20), as well as their feedback on the intervention. Participants gave informed written consent to participate in the study.

Participants were asked about the experience of living with complex conditions and concurrent aging to gain insight into how "work-is-done" in practice, and how people with multimorbidity are currently supported to focus on health goals/stay healthy. Participants also reviewed the 'CHAT&PLAN' intervention prototype. Semi-structured questions were used to explore what participants liked, disliked and thought should be changed. The interview guide is available as a supplementary file (S1 File).

After interviews were conducted, initial thoughts and ideas were noted down by the interviewers as an early stage of analysis. The data were transcribed verbatim. Data were analysed using thematic analysis to assess participants' thoughts about the intervention content and inform potential changes. Initial codes were identified and highlighted factors considered pertinent to the design and implementation of the intervention. The generation of initial codes was primarily done by one researcher (TC) with occasional cross-checking to independent coding by a second researcher (AY). Coding was discussed by members of the study team (TC, AY and JB) and developed into themes.

## Results

### Planning phase of developing the 'CHAT&PLAN' intervention

A rapid scoping review of the literature established that older adults LWBC differ in terms of functional status, cognition and comorbidity [31] and many have a number of conditions which affect their cognitive and physical functioning [4,32–38]. Older adults with multimorbidity may struggle to access information, emotional and practical support [1,8,14,39,40]. Crucially, the scoping search suggested many older people are likely to have untapped assets and resilience which, if deployed, could help them to better manage their health [32,41]. Healthcare professionals often report insufficient knowledge and skills to support older people with complex conditions [42]. Training and education may be required to encourage 'buy in' and to facilitate a shared understanding of purpose [43]. NICE guidelines for approaches to care that takes account of multimorbidity focus on how an individual's health conditions and their treatments interact and how this may impact quality of life (QoL). Guidelines advocate attending to the person's individual needs, preferences for treatments, health priorities, lifestyle and

goals. HCPs are encouraged to consider ways to improve QoL by reducing treatment burden, adverse events, unplanned care and fragmentation of care [44]. HCPs and the individual should agree a personalised management plan that incorporates goals and plans for future care and outlines who is responsible for care coordination [44]. Disease and treatment burden should be recognised, but goals, values and priorities should also be identified. These may include lengthening of life, maintenance of independence, taking part in valued activities, preventing specific adverse outcomes or reducing side effects of medicines, and reducing treatment burden [44]. There is a need for assessment of individual difficulties and variation in self-efficacy to self-manage so that support can be tailored appropriately according to level of need [45,46]. Some studies have described a need to enhance communication and establish a means of managing complex, fragmented care, in alignment with health service users' priorities [46,47].

The systematic review and synthesis of qualitative studies revealed older adults living with cancer and multimorbidity value autonomy and independent living as a key feature of quality of life [28]. Health conditions that had the greatest negative impact on independent living were prioritized. Often, a key driver of engagement with self-management 'work' was whether or not the healthcare practices were seen to interfere with QoL and/or aligned with their understanding of their health and symptoms. People were reluctant to burden others in their social network with help seeking. Healthcare services' role in supporting self-management was considered as peripheral to people's experience of daily living. Lack of time and difficulties in establishing a rapport with HCPs in clinical consultations interfered with trust being established. More responsive health care that aligns with individual priorities and preferences may result in improved health outcomes for this group. Older adults LWBC are often actively prioritising their own values and autonomy, but these actions may not align with formal service provision or HCP expectations of the patient's self-management role [28].

Based on findings from the reviews and from our research team, including PPI members, we developed an insight into the experiences of the healthcare providers and recipients who were the target audience for the 'CHAT&PLAN' intervention. The guiding principles were continuously and iteratively refined as new information emerged e.g., from the behavioural analysis and qualitative interviews. The finalised 'CHAT&PLAN' guiding principles are outlined in Table 1. This table summarises findings from the reviews, which also demonstrates how they were used to develop intervention guiding principles. We link these guiding principles to the aims outlined in the logic model, demonstrating how they informed the intervention development and helped us to identify key context-specific behavioural issues to be addressed.

Relevant evidence from scoping reviews, team expertise and qualitative interviews was tabulated, and were then mapped onto existing theory. This allowed clear description of the intervention processes, including the behaviours to be targeted and strategies to deliver these functions. A key aim of the intervention was to reduce the workload of cognitive and practical tasks for older adults with multiple conditions in order to increase their capacity to self-manage their health [20]. Theories of Health Psychology & behaviour change were employed to address the finding from both the rapid scoping review of the literature and synthesis of qualitative studies that psychological factors often shape individuals' overall response to health-related work, self-management and cumulative complexity. Subjective beliefs about (and attitudes towards) complexity were key drivers of behaviour, often more significant than the influence of objective patient workload and capacity. We drew on a range of psychological and sociological theories as outlined in Table 2.

Once developed, the 'CHAT&PLAN' targeted eight core behaviours based on the guiding principles and theoretical analysis (see Fig 2): initial 'checking in' with the service user and

**Table 1. Guiding principles summarising key intervention design needs and objectives.**

| Literature review findings | Intervention design objectives | Key features: | Link to logic model aims |
|---|---|---|---|
| • Older people may be less likely to engage in conversation with HCP, more likely to view HCP as "authority figure" [10,43,44]<br>• HCPs likely to underestimate or make assumptions about older people's healthcare preferences and priorities[16,40]<br>• Features of old age and some diseases can sometimes complicate communication [16,43,45] | • To ensure that health service users are given the opportunity to voice their opinions and concerns.<br>• To ensure that these are taken on board and acknowledged by the HCP | • Effective communication system to facilitate coordinated care and informed decision making.<br>• health service user given space and time to express views<br>• HCP encouraged to actively listen to health service users<br>• HCP and health service users work collaboratively develop goals | • To create a 'safe' goal-directed conversation where service-users can express their needs, concerns and values<br>• To ensure that service-users are given the opportunity to voice their opinions and concerns. |
| • Many older people have comorbidities and limitations which affect their cognitive and physical functioning [6,46–52].<br>• Many older people have untapped assets and resilience which help them to better manage their health [46,53].<br>• "Work" of multimorbidity: the burden of treatment, illness vs. capacity to manage/cope [19,20,44]<br>• Receive care from providers that is focused on specific diseases, which can be burdensome and fragmented [4,19] | • To encourage HCPs to ask health service users about how they are managing their health in general, taking a person-centred rather than disease centred approach | • Ensure a dialogue about cancer treatment if relevant, and address unmet physical, psychological and social support needs.<br>• Enables health service users to identify key health-related burdens/issues<br>• Ensures HCP is aware of any imbalances between workload/capacity | • To promote a person-centred rather than disease centred approach<br>• To prioritise values identified by the service-users |
| • Older people value a range of outcomes beyond survival [19,54,55]<br>• Older adults report over-emphasis on medical aspects of health- do not always get opportunity to express concerns, values and priorities [1,5,8,46,56] | • To prioritise values identified by the health service users<br><br>• To engage in a structured goal/value-focused conversation | • Health service users asked to outline their priorities<br><br>• Collaborative focus on goal-setting based on priorities set out by health service users | To ensure that opinions and concerns are taken on board and acknowledged by the healthcare provider<br>• To prioritise values identified by the service-users |
| • Older adults may struggle to access information, emotional support and practical support [1,10,16,40,57].<br>• low health literacy or self-efficacy may lead to a lack of confidence, or understanding [58,59].<br>• Factors such as cognitive impairment, social support needs and caring responsibilities are also likely be relevant [58]. | • To ensure that health service users feel enabled and equipped to cope with issues that are a priority to them | • Provision of structured guidance on how to set goals, make an action plan<br>• Goal setting identifies key aspects of achieving goals<br>• Tool designed to establish clear referral pathways for health service users (including voluntary sector agencies, social services, and specialist teams) | • To enhance capacity to engage in healthcare-related work |
| • Fragmented care: often not clear who takes responsibility for health service user needs [55,60–62] | • To negotiate roles and delegate tasks between health service users and HCP to remove ambiguity about role | • Discussion and clarification of priorities of both HCP and health service users<br>• Task delegation<br>• Duties/Responsibilities outlined and agreed | • To identify opportunities to reduce health-related workload |
| • At different stages, different conditions may be exacerbated or come to the fore. Approach needed that recognises and acknowledges 'flux' in capacity and burden [20,61,62]. | • To review goals and adapt goals as priorities change | • Follow up assessments undertaken at defined points to identify and address changes in need.<br>• Incorporate adjustments to address additional needs. | |

identifying 'what matters most' to them; active listening'; identifying and discussing service user priorities; linking patient priorities to health-related objectives; agreeing a goal; creating an action plan; agreeing responsibilities; negotiating roles and creating a sense of shared-responsibility. We drew on existing evidence relating to behavioural change techniques to identify those techniques most frequently used and appropriate to meeting the aims of the intervention [69,70]. These included identification of current skill set, problem solving, information provision, instruction and demonstration, coping planning, goal-elicitation, action planning, decision making, relapse prevention, and goal reviewing.

**Table 2. Outline of theories employed in intervention planning process.**

| Theory | Theory aims | Intervention aims associated with theoretical constructs |
|---|---|---|
| Burden of Treatment Theory [20] | To understand how capacity interacts with the work that stems from healthcare. | To assess of health-related workload of service user and consideration of coping with demands of burdens of illness/treatment |
| Shippee's cumulative complexity model (CCM) [19] | To explore how confounding factors at an individual level may accumulate due to multimorbidity [2] and how poor outcomes are likely to be derived from an imbalance between the patient workload of demands and patient capacity [2–4]. | To enhance capability to perform healthcare-related work and self-management by assessing the load of cognitive and practical tasks delegated to the health service user, so it does not become overwhelming.<br>To increase capacity to adapt to challenges associated with self-management<br>To empower health service users to renegotiate roles and responsibilities relating to their healthcare. In turn, this may increase capacity to manage conditions effectively |
| Cognitive authority theory [44] | To outline negotiation processes in which people manage important relational aspects of inequalities in power and expertise, particularly relating to the management of long-term conditions. | To attend to the lived experience of service users and consider how this impacts capacity self-manage.<br>To facilitate "experienced control" so that health-related workload is perceived as "doable". |
| Self-Determination Theory (SDT) [63] | to support patient autonomy in order to optimise their functioning Proposes that individuals seek supportive relationships in which their emotions and beliefs are respected (relatedness) [63]. Complements the use of motivational interviewing (MI) as it describes a process of self-motivated change [64]. | To employ strategies promoted in motivational interviewing, such as voicing empathy, exploring incongruity between current and goal behaviours, supporting self-efficacy and managing resistance [65].<br>To use active listening strategies to build a supportive dialogue that can lead to exploration of the issues at hand [66]. |
| Participative goal setting [67]. | To facilitate creation of goal acceptance and commitment goal acceptance: initial agreement with a goal. commitment: one's attachment to or determination to reach a goal | To promote proactive participation in the consultation, with strategies to facilitate participative goal setting and ensuring that both service-user & clinician are involved in the decision making process |
| Gollwitzer's concept of implementation intentions [68] | To promote the initiation of goal-directed actions.<br>Forming a goal intention can create commitment to achieve the end state. An implementation intention is a self-regulatory strategy that links a specific goal-directed behaviour to an anticipated future state (opportunity).<br>An implementation intention is a commitment to perform the specified goal-behaviour when the opportunity is encountered. (" I intend to do $A$ when situation $X$ is encountered.") | To create a specific action plan outlining when, where and how the goal intention will be transformed into action |

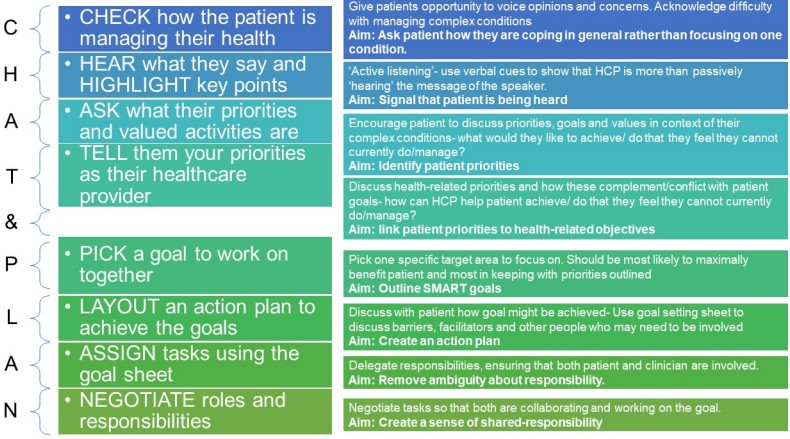

**Fig 2. Original prototype of the CHAT&PLAN.**

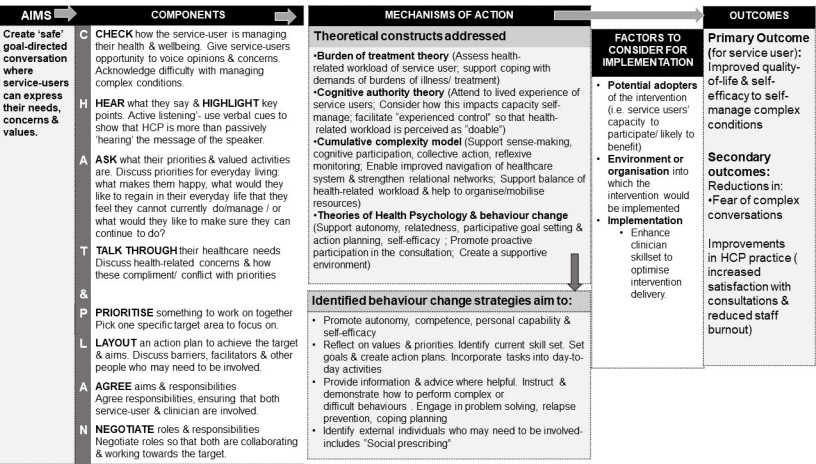

**Fig 3. Logic model of the CHAT&PLAN intervention.**

The logic model (Fig 3) illustrates the hypothesised mechanisms of action of the 'CHAT&-PLAN' and outlines the programme theory.

## Optimisation phase of developing the 'CHAT&PLAN' intervention

Demographic information for each participant can be seen in Table 3. Findings from the interviews informed iterations and modifications of the guiding principles and the behavioural analysis. They also identified required intervention changes. Table 4. provides examples of qualitative interview data that informed the key points raised below. Table 5. provides an overview of changes made, including examples of participant feedback. Fig 4. shows a modified version of the CHAT&PLAN, including recommended prompts and advice suggested by interviewees.

Some of the key points to emerge from the interviews are outlined below, and supporting quotes can be found in Table 4.

**Non-clinical, shared- approach to care viewed positively, if appropriate for the service-user's needs and if it does not create extra work for those receiving care or for the HCPs.** A non-clinical approach focusing on health service users' priorities was seen as a favourable feature and considered as a way of facilitating holistic care. HCPs and older adults liked the shared-nature of the plan, which was seen as empowering individuals whilst ensuring they were supported. Crucially, most HCPs felt very strongly that this should not create extra work for those receiving care, given the amount of health-related work that is already expected of those with multimorbidity. Conversely, some older adults were concerned that the plan would create additional work for HCPs, reflecting on existing pressures on staff and an increasingly overburdened healthcare service. The use of the tool must be appropriate for the individual's needs, abilities, and priorities. They noted that it may be confusing and difficult for those with difficulties in cognitive impairment, lower health literacy, higher illness burden or those with more complex issues. HCPs noted that the approach would not work for everyone and some suggested that the CHAT&PLAN is used with those identified as most likely to benefit. HCPs emphasised the importance of how the CHAT&PLAN is discussed with the person receiving care. They should be interested in participating, know why this approach is being used and understand how it might be different from a more traditional consultation.

**The structure may help to routinise and formalise practice that may already happen if staff have the support, time, resources and skills to do so.** Participants often liked the

**Table 3. Demographic information of participants in qualitative interviews.**

| Healthcare professionals | | |
|---|---|---|
| **Participant ID** | **Gender** | **Job Title (HCP)** |
| 102 | Female | Macmillan Allied Health Professional (Cancer Rehabilitation Lead) |
| 103 | Male | Consultant (Geriatric Medicine) |
| 104 | Female | Nurse specialist (Lymphoma) |
| 105 | Female | Consultant Nurse (Frailty) |
| 106 | Female | Nurse specialist (respiratory) |
| 107 | Female | Advanced Clinical Practitioner (Frailty) |
| 108 | Female | Support Worker |
| 109 | Female | Nurse (Long Term Conditions Lead) |
| 110 | Male | Consultant (haematologist) |
| 111 | Male | Nurse (Frailty & Older Persons Rapid Assessment Unit) |
| 112 | Female | Consultant (Pain) |
| 113 | Female | Nurse Consultant (Older Person's Mental Health) |
| 114 | Female | Research Nurse |
| 115 | Female | Support Worker (Neuroendocrine Tumour, Upper GI & Anal Cancer) |
| 116 | Female | Clinical Nurse Specialist (Upper GI Oncology) |
| 117 | Female | Cancer Support Worker |
| 118 | Male | Oncology nurse |
| 119 | Female | Clinical Nurse Specialist (Head and Neck, and Thyroid) |
| 120 | Female | Cancer Nursing Management |
| 121 | Female | Charity Ambassador for local Cancer Support Centre |

| **Healthcare Recipients** | | | | | | | |
|---|---|---|---|---|---|---|---|
| **Participant (PPT) ID** | **Gender** | **Age** | **Education** | **Cancer Type** | **Cancer Treatment** | **Completed Treatment** | **Other long-term conditions** |
| 101 | Male | 76 | Trade/technical/ vocational training | colon | surgery | 18.09.2012 | COPD; Asthma; Sleep Apnoea; Prostate; Atrial Fibrillation; stomach ulcer |
| 122 | Female | 78 | Trade/technical/ vocational training | colon | surgery | | arrhythmia/atrial fibrillation, rheumatoid arthritis, fluid retention, walking problems |
| 123 | Female | 83 | Trade/technical/ vocational training | rectal | surgery | 4.07.2017 | Overactive bladder, atrial fibrillation, other arthritis, previous history of minor myocardial infarction, heart failure, spinal compression (in neck brace), osteoporosis, torn shoulder ligaments (bilateral), falls (fell day prior to interview- bruising) |
| 124 | Female | 80 | Trade/technical/ vocational training | rectal | radiotherapy, chemo, surgery | 14.11.2017 | high blood pressure, underactive thyroid, |
| 126 | Female | 79 | Secondary school/ college | bowel | surgery | 24/05/2017 | Arthritis, high blood pressure/hypertension, Diverticulitis, optic rotatory dispersion. |
| 127 | Male | 69 | Secondary school/ college | bowel | surgery | 20/04/2018 | asthma/COPD, chest pain, neuropathy, enlarged organs |
| 128 | Female | 88 | Secondary school/ college | colon | surgery | Mar-18 | Arrhythmia/irregular heartbeat (e.g. AF or atrial fibrillation)/osteoarthritis, Diabetes, High blood pressure, underactive thyroid |
| 129 | Female | 77 | Secondary school/ college | endometrial | surgery (brachytherapy) | Jul-18 | High blood pressure or hypertension |

| **Caregivers** | | | | | |
|---|---|---|---|---|---|
| **Participant (PPT) ID** | **Gender** | **Age** | **Education** | **Relationship to participant** | **Long-term health conditions** |
| 125 | Male | 82 | pharmacy college | Spouse of participant 124 | Arrhythmia |
| 130 | Male | 81 | Secondary school/college | Spouse of participant 129 | Other Arthritis (e.g. osteoarthritis, psoriatic arthritis) |

**Table 4. Qualitative interview data.**

| Finding | Sample Supporting Quotes from Healthcare Providers | Sample Supporting Quotes from Healthcare Recipients and caregivers |
|---|---|---|
| Non-clinical, shared- approach to care viewed positively, if appropriate for the service-user's needs and if it does not create extra work for those receiving care or for the HCPs | *A lot of these patients by the very nature that they're complex, multiple morbid, have got other plans. You know, whether it's a care plan from the district nurses, a care plan from the dietician or speaking language or community matron, or GP or existing advance care plan. So, they've got lots of plans. They've usually got a folder if they are accessed by the caregivers or external sort of social services. So, it's all that information. It's all out there. So, this is just another one. So, it needs to fit in with the existing patient management.–Consultant (Geriatric Medicine)* | *It probably would be quite helpful-* PPT 124 (female age 80)<br>*I think, it's the more personal touch, to be honest. Yeah. Not: "oh, you've got five minutes. You've had four. Your time's up" sort of thing.–PPT 122 (female, age 78)* |
| The structure may help to routinise and formalise practice that may already happen if staff have the support, time, resources and skills to do so. | *I think, that we should be doing it and it's already there but the structure is just one of those things that would really support people that were not as confident or didn't do it as much.–Nurse (Frailty And Older Persons Rapid Assessment Unit)*<br>*And obviously, having a structure is good. I think, particularly for newer staff it will make them think as well. As well as—you know, if they're working through it with a patient. It's also setting off some thought processes for them. I think, it's something that largely we do but probably in an unstructured way. So, again, for people—generally, when you're kind of busy and you're flat out to have a structure to it is really good. For people who are more junior and new to CNS roles, you know, getting them to kind of think outside of the box and not just be: "this is the cancer I deal with. This is my little tunnel and I don't step out of it" will be really good.–CNS (Upper GI Oncology)* | *I just feel I'm just a number around [at the GP surgery]. That it doesn't—that I don't really matter anyway-* PPT 122 (female, age 78) |
| Concerns related to time and staffing, risk of creating another "tick-box" exercise for staff, and challenges associated with fragmented care | *I think, staff will always see it as a very time consuming. I see it as frontloading. I see it as: "get this right in the first place you save, you know, a lot of time later". You get a better relationship with your patient early on. But, I think, for staff out at the coalface they'd have difficulty crafting that care plan and be they will always say they haven't got time to.- Consultant Nurse (Frailty)*<br>*Just thinking about it you've then got one document and one assessment and then it's then talking about a target sheet. So, then that's another piece of paper. And, I'm not entirely convinced how much people want all of these different assessments. I think, a lot of patients fill them in because they feel obliged to. Not because they want to. And, therefore, are we actually using a lot of time and energy and resource doing things that people don't want? And it all just feels a bit of a waste of time when you know that you've kind of done five hours unpaid overtime that week just to get things that absolutely needed doing done because you were doing this nice new fancy assessment to make the boss happy. And it's starting just to feel a little bit like assessment for assessment sake rather than it actually being what patients need and want.—CNS (Upper GI Oncology)* | *To me, it would be the specialist but, having said that, they don't have time because they've got so much to do.–PPT 126 (female, age 79)*<br>*If they've got time. That's what I found in there. Poor girl—nurses would turn—you know, turn from one place to the other. I can't fault them for what they did. You know, they were brilliant. But they just never had time to talk to you or to do anything.- PPT 128 (female, age 88)* |
| The tool would work best if it was linked with something that was already happening in practice, preferably outside of the hospital context | *It depends on the patient. I think, you could use all three of those methods but, I think, it would need to be the right method for the patient.-. I think, if you can do a good chat and plan, you should be able to do it anywhere.- Consultant Nurse (Frailty)*<br>*It's not a set intervention. You have to be ready to step out of the intervention if someone presents with something challenging clinically and that can be physical as well. A couple of people did get readmitted to hospital and became more unwell. So, you kind of have to start the cycle again is what we find. You have to recycle it. So, I think, it is a cycle and a circular process.—Nurse (Long Term Conditions)* | *In a clinic because most GP surgeries have spare rooms. And usually the GP surgery is close to where the person lives. Reasonably. But if the person's unable to leave the house well, obviously, it would have to be at home. But other than that I would say at the GP surgery because it—it's a psychological thing. . .- PPT 101 (male age 76)*<br>*Face-to-face is better. Facetime I can cope with perfectly alright. 'Phone because I'm like deaf as well and then you've got to have a very good line, a very good voice and—And, I don't mean to be but I—I'm not rac—but the minute I get these very heavy accents, like I say, on the 'phone, I'm not so good.- PPT 128 (female, age 88)* |

*(Continued)*

**Table 4.** (Continued)

| Finding | Sample Supporting Quotes from Healthcare Providers | Sample Supporting Quotes from Healthcare Recipients and caregivers |
| --- | --- | --- |
| Initial priority should be service-user safety; staff must know limits of their knowledge and when to refer | *You do have to check how—that the patients are safe and, you know, check that they're safe with the long-term conditions. There's nothing acute going on.*–Nurse specialist (Respiratory)<br>*This support worker role you have to be—you have to watch this role because you've got quite a lot of responsibility in a way. And, I think, if you didn't have the background and the support you could get lost. Or you could end up in a situation. I am so, so clear to say: "listen, I'm fully here to support you but this is something that is not within my level of expertise. But, I'm going to go and call the nurse and she's gonna come down and see you". Or: "she will 'phone you at home". You've got to do your boundaries.*—Cancer Support Worker (PPT 117) | *Providing the person knows what they're talking about. . . Multiple skills. Don't have to know the answers. All they need to do is understand what the patient's talking about and half the answers are common sense. There's always a chance say: "well, OK, I'll make a note of that. I'll speak to the GP and we'll get back to ya within, you know, a couple of days. I'll get them to give you a ring if they can". That's all you need. But they must understand what the—I know it's difficult, somebody to understand everything but it's just a little bit about everything so they can relate to the patient that's talking to them. And, I think, that's the answer. . .*- PPT 101 (male age 76) |
| Anyone with the appropriate skills could potentially deliver the intervention, but training may be required to ensure correct utilisation of the tool and self-efficacy to deliver to the intervention | *Anyone with the appropriate skills to do so. So, anyone directly involved with patients. They're managing patients. So, it could be anyone from sort of medical to nursing to community teams as well—as long as they've got experience with managing younger/older patients with multiple issues.*- Consultant (Geriatric Medicine)<br>*Definitely need to train them. They need how to listen to the question. How to goal set. How to action plan. I'd say all that needs training. And we've all got it to a certain extent but to use that effectively you'd have to train somebody and particularly in the putting peoples' wishes first part. And capacity. Consent. How to listen. How to get somebody's story if they don't tell their story like I'm telling you. So, somebody who's got a cognitive problem. Speech problem. Hearing problem.*—Advanced Clinical Practitioner (Frailty) | *The doctors. They listen but, I think, they block some of it out as you're talking to them. That they knows what to pick out, I think. They're—where they're used to it so much. Do you know what I mean? They knows the important bits. They can pick them out. But, as I say, (Cancer CNS) she was sort of—listen all the time. She's—give you time to reply, if you know what I mean. Whereas, some of them before you can say what you want to say, they're asking you the next question. But she was as good as gold.*- PPT 127 (male, age 69) |

structure offered by the CHAT&PLAN, and the acronym, which was considered clear and self-explanatory. HCPs may already engage in person-centred care if they have the support, time, resources and skills to do so, though some noted a lack of a structured approach. Individual differences between health professionals in personality and empathy were also mentioned. Most participants believed that the CHAT&PLAN would be useful to prompt staff to engage in person-centred care and would be particularly helpful for new staff. Some believed the structured approach could help to standardise practice. However, others suggested that a structured approach may distract from the consultation, as it may be difficult to remember the steps.

**Concerns related to time and staffing, risk of creating another "tick-box" exercise for staff, and challenges associated with fragmented care.** Time and staffing to accommodate an extra consultation were raised as potential barriers to implementation. HCPs described variation in how existing programmes were delivered as recommended, often due to a lack of available resources. Some warned against creating another "tick-box" for staff, arguing that such exercises often divert attention from meeting immediate care needs. Individuals questioned the added benefit of the CHAT&PLAN approach and how it might link in with existing programmes of work. Many spoke about the issues relating to fragmented care, noting that healthcare often happens in silos and information is often not efficiently transferred across teams. This made it difficult for HCPs to access information pertaining to those receiving care. However, by accessing online healthcare records, some believed that CHAT&PLAN could facilitate integration of care across teams by placing the health service users' own priorities at the centre.

**Table 5. Overview of the changes made, including examples of participant feedback.**

| Original text | Quote | Change made |
|---|---|---|
| **C**HECK how the patient is managing their health | *I can see this but maybe just changing the words like (managing their health)... Managing their life or something like that instead.–Cancer Support Worker (Neuroendocrine Tumour, Upper GI & Anal)* | **C**HECK how the health service user is managing their health and wellbeing |
| TELL them your priorities as their healthcare provider | *I think, things like: "tell them—**tell them your priorities**". I'm not sure: "tell" is the right word. I think, just making sure that the language is slightly tweaked to ensure that it's really patient facing and takes the sort of wagging finger element out of things-* Consultant Nurse (Frailty)<br>*The only thing I thought that was a little bit that I would sort of discuss would be the **"T"**. So: **"telling them"**. Maybe rewording that because it sounds like—I don't know how to put it really. Rather than telling them it should be about sort of a two-way conversation.–* Nurse Specialist (respiratory) | **T**ALK THROUGH their healthcare needs |
| PICK a goal to work on together | *I don't need goals. Because if some health care professionals they'll say: "oh, we need to do this and we need to do that". Shutter drops. But—and I know a few other people the same. They've had it for a long time. They don't need to be told or given goals to go for. They know exactly where they're going. But with people with a newly diagnosed, yes. It gives them something to go for. But to us old ones it's phew. No (laughter) because we've got all our goals. We know our limitations. We know what we can do and what we can't do- PPT 101 (male age 76)* | **P**RIORITISE **something** to work on together |
| Layout an action plan to achieve the goals | *The idea, as you say, to find out and sort of achieve a goals is good enough but, as I say, it —I've really—haven't got nothing I want to achieve. As long as I stays healthy- PPT 127 (male, age 69)* | **L**AYOUT an action plan to meet the **target and aims** |
| Assign tasks using the goal sheet | *Don't like: **"tasks"**. I would like it to be a little bit more patient orientated.-* Consultant Nurse (Frailty)<br>***Assigned tasks** it just probably needs wording so that we're not kind of tasking a patient with something but, I get what the concept is. And, again, this is probably just look at the light negotiating. So, we probably couldn't do assigning but there might be agree... That would be a good one wouldn't it? Sort of **agree some tasks together** and say who's going to be doing what which is essentially what we sort of doing at the end as well.-* Nurse (Long Term Conditions Lead)<br>*"**Assign tasks using the target sheet**". Task again. It takes me back to—it's taking me back to, you know, 20 years ago in nursing where we were task orientated. And are we asking people to do tasks? They are not necessarily tasks there just, I think, responsibilities.-* Cancer Nursing Management | Changed "tasks" and "assign"<br>**A**GREE aims and responsibilities using the target sheet |

**The tool would work best if it was linked with something that was already happening in practice, preferably outside of the hospital context.** CHAT&PLAN is likely to be more feasible if the consultation is embedded in routine practice, linked in with something that was

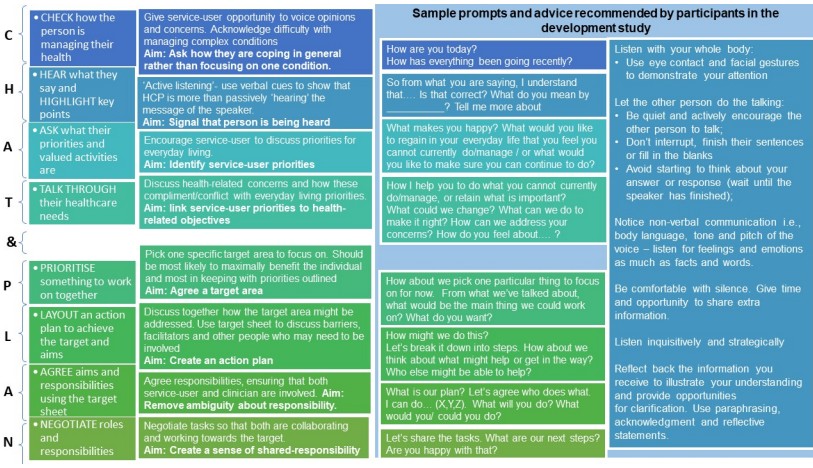

**Fig 4. Modified prototype of the CHAT&PLAN.**

already happening. It was proposed that CHAT&PLAN be used alongside the development of care plans or treatment summaries. Most agreed that CHAT&PLAN lends itself to multiple points on the care trajectory, but likely to have a different focus at various transition points. However, some preferred the idea of having a conversation as early as possible to gain insight into an individual's priorities from the outset. Conversely, others thought it would be better placed after treatment, given the self-management focus. Many believed it would be best done outside hospital, perhaps in a general practice setting. Generally, both HCPs and older adults concluded that location and mode of delivery must work for those it was designed to help and therefore should be chosen by participants. HCPs emphasised it should be an ongoing, flexible and fluid intervention, working with the individual as their needs change.

**Initial priority should be service-user safety; staff must know limits of their knowledge and when to refer.** It was emphasised that at the outset of any consultation, HCPs must ensure that people are safe and that there are no immediate or acute issues. Staff must know what to look for (potential problems, symptoms or conditions that might be linked, things that might be issues later etc.). It was perceived as key that HCPs know their boundaries and when they should refer to other teams. There was often discussion around relative merits of generalist versus specialist practitioners in working with older adults with multimorbidity. An alternative suggestion was to have the conversation led by a non-clinician who would link-in with the clinician or medical team. Support workers were mentioned as having many of the skills required to deliver the intervention, but may not have the advanced clinical skills needed to detect acute health needs or potential exacerbations of conditions. Therefore, a support worker-led consultation would need to occur alongside a routine consultation (e.g. after a cancer care review at a GP practice) or be supervised by a clinician for safety.

**Anyone with the appropriate skills could potentially deliver the intervention, but training may be required to ensure correct utilisation of the tool and self-efficacy to deliver to the intervention.** Generally, HCPs believed that anyone with the appropriate skills could potentially deliver the intervention. The key was to ensure that they had been trained to optimally utilise the CHAT&PLAN. Training would also help to standardise the way the CHAT&PLAN is used and build clinical confidence and self-efficacy. Staff should be supported by providing training, rehearsal, feedback, time, space and mentoring. HCPs frequently mentioned that motivational interviewing and health coaching skills would be helpful for the delivery of the intervention. Participants noted that there would be a need to consider people's understanding of their health and frequently discussed the importance of being able to refer or signpost to other services.

Active listening approaches were emphasised as a key skill by both HCPs and older adults. The majority of HCPs acknowledged it was often challenging to truly listen to what a person wants and to create an open environment for conversation. HCPs discussed individual differences in empathy, and noted that it would take experience to be able to frame the CHAT&PLAN as a conversation and to pick up on clues that the individual may need extra help. HCPs wanted to develop skills to be responsive and maintain a sustained focus on the care-recipient's priorities, especially in cases where their priorities might conflict with those of the care-recipient. Both HCPs and older adults raised concerns about the perceived open-ended nature of the conversation, stating it may be difficult for some HCPs to have a consultation that was led by health service users. There was concern about opening a "Pandora's box" of issues that the HCP would not be able to manage and it was deemed important to educate HCPs on how to help an individual to identify key priorities. Participants discussed the challenge of setting realistic, small, achievable goals that are important and meaningful to health service users. Some participants noted that goal setting is quite a novel skill for many HCPs, and one that could be quite difficult for them. Training in goal setting would help HCPs to be able to develop a plan

to achieve the goals, as well as managing expectations, goal reviewing, modification and evaluation.

**Modifications needed to optimise the feasibility of the intervention.** One of the aims of the study was to identify intervention modifications needed to maximise the success of the intervention. We explored what participants thought should be changed to improve intervention acceptability. Some participants expressed dissatisfaction with some of the terminology used. As an illustrative example, HCPs contended that goal-setting was becoming increasingly common in modern practice. However, there was some confusion amongst older adult participants about the concept of goals. Many of those who had long-term conditions expressed a belief that goals are for those who have a new condition, stating that their only goal was to stay healthy. There was a sense that these individuals would be less likely to engage in a conversation that was pitched as being a "goal-setting exercise". Therefore we explored other options such as the word "target" or "aims" which were considered more favourably. A key factor in achieving person engagement with the intervention would be how the purpose of CHAT&PLAN was explained to the person receiving care.

## Discussion

In this paper we have presented an overview of an evidence-, theory- and person-based development process for an intervention which aims to facilitate person-centred care and in turn, improve QoL in older adults living with multimorbidity. Design and planning were conducted by a multidisciplinary team and informed by theory, policy, guidelines and the findings of a systematic review of qualitative literature. It was refined and further developed with input from older adults, their caregivers and healthcare professionals. We have provided a detailed account of how and why the intervention took its current form, and how it is expected to work. In doing so, we have provided useful and transferable insights into the issues that are likely to be involved in delivering a complex intervention to deliver person-centred care to older adults.

This study provides empirical evidence that a conversation-based intervention to promote person-centred care may be acceptable and engaging for older adults with multimorbidity and healthcare professionals who work with them. To date there has been insufficient application of theory to understand and guide intervention development for care of those with multimorbidity [18]. Use of theory in the development process enabled us to develop an holistic approach to support those with multimorbidity, founded on the priorities and values of the individual, rather than on treating specific health conditions or symptoms [71]. The aim of CHAT&PLAN is to improve quality of life by enhancing patient capacity and self-efficacy to self-manage, as well as reducing treatment burden, adverse events and unplanned or fragmented care. This requires an adaptive approach that enables an individual to address their care needs (that may vary and fluctuate over time) in order to optimise wellbeing [31].

Vermunt et al [32] sought to evaluate studies on the effects of interventions that support collaborative goal setting or health priority setting compared to usual care for elderly people with a chronic health condition or multimorbidity [32]. The authors concluded that collaborative goal setting and/or priority setting is probably best when integrated in complex care interventions. The authors recommended that future research should determine the mix of essential elements in a multifactorial intervention to provide recommendations for daily practice [32]. Interestingly, our qualitative work evidenced discrepancies between older adults' and HCPs' interpretations of the meaning of particular health-related terms or language, which may make it difficult to productively engage in collaborative goal setting and/or priority setting. For example, the majority of HCPs thought that the word "goals" was suitable and

appropriate for the intervention, demonstrating an increased focus on goal-oriented approaches to care in recent years [15]. However, identification of goals may be more complicated than anticipated. Our findings indicated that older adults were confused about what the term "goals" meant for them, given that they had lived with their conditions for many years and had already adapted their lives to accommodate their conditions. Individuals reported they perceived "goals" as achievement-orientated, or something to be focused on when a new condition was diagnosed. The term "goal" was not perceived as reflecting a desire to *maintain* health or independence and was rejected by many of the older adults we spoke to. Some research has indicated that individuals with multimorbidity do not naturally share their goals with providers [33]. If those in receipt of care misinterpret the meaning of a question relating to goals (e.g. if they do not think they have any goals) then they may not receive adequate support to help them to meet their ongoing needs. Thus, while goal-setting is likely to be a helpful exercise, care must be taken in how the topic is approached and explained, so that appropriate goals or aims can be developed (e.g. explaining that a goal can be a short-term aim, or can be something small that the individual would like to change or maintain). The use of appropriate, meaningful language is important to ensuring engagement in healthcare. To reflect the feedback of older adult participants, we changed the terms used in the CHAT&PLAN tool. HCPs must be trained specifically in how to clarify the concept of goal-setting as a means of helping people to think about how they would like their future to look and how this might be made possible. It is key that HCPs demonstrate an understanding of the context of living with long-term conditions and the expertise held by the individuals living with those conditions [33].

Findings from a trial of a patient-centered intervention (the 3D intervention) was recently published [34]. The authors concluded that effectiveness of the intervention could be improved by further training of practice staff, promoting greater consistency and generalist training to promote confidence to support a variety of longer-term conditions. The study included 33 general practices (1546 patients) and found the intervention to improve management of multimorbidity did not result in a meaningful effect on patients' quality of life [34]. However, the study authors of the 3D intervention concluded that effectiveness of the intervention could be improved by further training of practice staff, promoting greater consistency and generalist training to promote confidence to support a variety of longer-term conditions. In particular, post-hoc process evaluations recommended further training for goal-setting as a key concept in patient-centred care, something our study identified. Another recommendation was that the intervention should have had less of a medical focus, and instead emphasize meeting social care needs through social prescribing and signposting to services within the community. In our extensive qualitative development work, one of our key findings related to the skillset and confidence of the healthcare professionals, and the potential for variations in practice between different HCPs. Similarly, the HCPs that we interviewed, also stated that they would like to develop skills such as goal-setting and signposting. Thus, our qualitative and iterative development process has enabled us to identify many potential implementation issues before the intervention is tested further, and in turn reduced research waste by identifying modifiable barriers to implementation [35].

The need to train healthcare professionals has been identified in previous reviews. For example, a Cochrane review reported on the benefits of personalised care planning, highlighting improvements in indicators of physical and psychological health status, and capability to self-manage conditions [17]. However, the authors concluded that it would probably require training for health professionals in how to elicit patients' goals and priorities, suggesting that investment in relevant training, support and system redesign could lead to better outcomes for people with long-term conditions [17]. Another Cochrane review [36] evaluated communication skills training (CST) in changing behaviour of healthcare professionals (HCPs) working

in cancer care. After the intervention HCPs were more likely to show empathy and to use open questions, and less likely to provide only facts. However, there was no improvement in communication skills, including eliciting concerns, clarifying and/or summarising information, and negotiation [36]. Drawing on our findings and existing evidence, prior to testing the efficacy of the CHAT&PLAN intervention on health service user outcomes, we must first be sure that HCPs are trained to effectively deliver and facilitate the intervention. This would include shared decision-making about goals with service users, as well as how to set and prioritise goals. This is likely to require more effective communication and coordination between HCPs [32,37].

Complex, novel interventions are notoriously difficult to integrate into practice [13], yet are more likely to be accepted when people are involved in the decisions and activities that affect them [38]. Our findings complement previous research indicating some clinicians are uncomfortable with the change in the relationship and power dynamic [39], or may perceive a collaborative approach as time-consuming and less willing to adopt such an approach [40]. Further, our work supports findings that health care professionals believe the care they provide is already person-centred, despite extensive evidence suggesting that this is often not the case [39,72]. The next step is to develop learning resources to underpin the development of skills to utilise the CHAT&PLAN in practice. This resource would help HCPs to establish how best to integrate the tool into practice.

## Strengths and limitations

We had initially aimed to recruit more participants in the patient groups. However, despite contacting a number of potentially eligible individuals, many chose not to participate. Unfortunately, we only recruited 2 caregivers. This is because we aimed to recruit caregivers identified by the older adult participants. The other participants did not identify a caregiver to participate in the study; four of these participants lived alone.

Difficulties in recruitment may be due to the nature of the conditions we wished to study and the busy health-related workload experienced by those with ill-health [73]. As well as health-status, other factors that could impact recruitment may relate to the ethnicity of participants. All of the participants who chose to participate in this study were white, which may be a reflection of the ethnic diversity in the local area (77.7% of residents recorded their ethnicity as White British in the 2011 Census [41]). However, efforts should be made to gain an insight into the experiences of those of different backgrounds, and to use different methods of approach to encourage participation in research [42]. Moreover, we did not ask specific questions about the socio-economic status of participants, yet it was evident that some were affluent as indicated in conversations about private healthcare and paying for support such as gardeners or cleaners. However, other participants did address financial challenges relating to paying for parking at the hospital and purchasing items such as sanitary pads.

It is worth noting that we did gain an insight into experiences of living with multimorbidity by drawing on the findings of previous research in our qualitative review and synthesis. The findings of the synthesis reflected the views of more than 960 patients and 52 family caregivers. The findings of the review were largely echoed by participants in our study. Further, while it was important to explore the acceptability of the tool in patient groups, our primary focus was on healthcare providers as they were the ones that would use the intervention. These individuals offered a more detailed insight into how the CHAT&PLAN would be implemented into routine practice.

CHAT&PLAN was originally designed for use in settings for older people who have cancer alongside multimorbidity. However, as the research progressed our findings suggested it was transferable for use by teams in other populations. Our qualitative review of the literature found that health conditions perceived to have the greatest negative impact on independent

living were prioritised by health service users, and so, for many individuals, previous experiences of cancer could assume a low priority. Similarly, in our qualitative study, many participants questioned the limited focus of the study on older adults with cancer and other conditions, arguing the tool could potentially be beneficial for anyone with multimorbidity. Therefore, in future studies, we will test the efficacy of the tool in a broader population.

## Conclusion

This paper provides a detailed description of a methodological approach to intervention planning and optimisation for a person-based intervention.

The study has elicited barriers and facilitators to implementation of the intervention and behaviour change [22]. We will use these findings in planning future research to test the effectiveness of the intervention to improve QoL in those with multimorbidity. In particular, this work demonstrated a need for training to enhance HCP self-efficacy and develop skills required by HCPs to optimally use CHAT&PLAN in practice. Thus, our next step is to systematically develop and test a learning resource to accompany introduction of CHAT&PLAN. This study has provided evidence that a person-centred intervention appears acceptable by healthcare professionals and appealing to older adults.

## Supporting information

**S1 File. MOCs interview schedule- caregivers- version 1 23102018.**
(DOCX)

**S2 File. MOCs interview schedule- patients- version 1 23102018.**
(DOCX)

**S3 File. MOCs interview schedule-HSCP—version 1 23102018.**
(DOCX)

## Author Contributions

**Conceptualization:** Teresa K. Corbett, Amanda Cummings, Kellyn Lee, Lynn Calman, Naomi Farrington, Lucy Lewis, Hilary Boddington, Theresa Wiseman, Claire Foster, Jackie Bridges.

**Data curation:** Teresa K. Corbett, Vicky Fenerty, Alexandra Young, Alison Richardson, Claire Foster, Jackie Bridges.

**Formal analysis:** Teresa K. Corbett, Amanda Cummings, Kellyn Lee, Vicky Fenerty, Alexandra Young, Alison Richardson, Claire Foster, Jackie Bridges.

**Funding acquisition:** Jackie Bridges.

**Investigation:** Teresa K. Corbett, Amanda Cummings, Alexandra Young, Hilary Boddington, Theresa Wiseman, Alison Richardson, Claire Foster, Jackie Bridges.

**Methodology:** Teresa K. Corbett, Amanda Cummings, Kellyn Lee, Lynn Calman, Vicky Fenerty, Naomi Farrington, Lucy Lewis, Alexandra Young, Alison Richardson, Claire Foster, Jackie Bridges.

**Project administration:** Teresa K. Corbett, Jackie Bridges.

**Resources:** Teresa K. Corbett, Lynn Calman, Naomi Farrington, Alison Richardson, Claire Foster, Jackie Bridges.

**Supervision:** Vicky Fenerty, Alison Richardson, Claire Foster, Jackie Bridges.

**Validation:** Jackie Bridges.

**Visualization:** Amanda Cummings.

**Writing – original draft:** Teresa K. Corbett.

**Writing – review & editing:** Teresa K. Corbett, Amanda Cummings, Kellyn Lee, Lynn Calman, Vicky Fenerty, Naomi Farrington, Lucy Lewis, Alexandra Young, Hilary Boddington, Theresa Wiseman, Alison Richardson, Claire Foster, Jackie Bridges.

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
