## [Decision Letter · Decision Letter 0]

17 Jun 2020

PONE-D-20-07914

Planning and optimising CHAT&PLAN: a conversation-based intervention to promote person-centred care for older people living with multimorbidity

PLOS ONE

Dear Dr. Corbett,

Thank you for submitting your manuscript to PLOS ONE. After careful consideration, we feel that it has merit but does not fully meet PLOS ONE’s publication criteria as it currently stands. Therefore, we invite you to submit a revised version of the manuscript that addresses the points raised during the review process.

This is an interesting and timely paper, but the reviewer has made several suggestions regarding the structure of the manuscript to improve the clarity of the work.  I invite you to consider these points in your resubmission.

We look forward to receiving your revised manuscript.

Kind regards,

Adam Todd, PhD

Academic Editor

PLOS ONE

Journal Requirements:

2.  Please provide additional details regarding participant consent. In the ethics statement in the Methods and online submission information, please ensure that you have specified (1) whether consent was informed and (2) what type you obtained (for instance, written or verbal, and if verbal, how it was documented and witnessed). If your study included minors, state whether you obtained consent from parents or guardians. If the need for consent was waived by the ethics committee, please include this information."

3. Please include a copy of the interview guide used in the study, in both the original language and English, as Supporting Information, or include a citation if it has been published previously.

Reviewers' comments:

Reviewer's Responses to Questions

**Comments to the Author**

1. Is the manuscript technically sound, and do the data support the conclusions?

Reviewer #1: Partly

2. Has the statistical analysis been performed appropriately and rigorously? 

Reviewer #1: N/A

3. Have the authors made all data underlying the findings in their manuscript fully available?

Reviewer #1: Yes

4. Is the manuscript presented in an intelligible fashion and written in standard English?

Reviewer #1: No

5. Review Comments to the Author

Reviewer #1: This is a really interesting and timely paper on the design of an intervention to support person centered care. A number of steps were taken to develop the intervention including two literature reviews, integration of theories and interviews. Due to the number of stages and steps it is a bit confusing how all of the pieces fit together.

- The introduction is clear, compelling and well-written

- As the paper goes on it becomes very confusing given all of the steps and stages that the authors undertook (I appreciate that this is hard to describe in one paper) but a number of modifications can be made to enhance clarity. I describe below...

- The research questions need to be upfront and clear (not tucked away into a table).

- All of the figures are blurry and hard to read (these are essential to the paper and need to be fixed)

The logic model is unreadable even when I downloaded it as a separate file. It does seem quite text heavy though. Can a more user friendly, less text dense version be prepared? It seems that this is central to the paper and more detail is needed on how it was put together and how the patient partners on the team helped to shape it.

It appears that a scoping review was conducted as well as an 'in-depth' formal qualitative literature review (submitted for publication)- please describe this method in a few lines. For the qualitative lit review, references should be added into that paragraph as well.

In the discussion there is mention of a systematic review (typo?). Was the scoping review published too?

Top of page 6- first sentence has typos."...insufficient of knowledge and skills..." as well as further down "...aim to agree a personalised management plan..."

Page 7- paragraph two "a key finding from both reviews"- is a very dense paragraph. The authors then go on to discuss how self determination theory shaped the intervention. There are two other theories mentioned earlier which are not defined and not linked to the intervention. Perhaps this paragraph can be turned into another table (column 1 - theory; column 2- definition of the theory and its core constructs/assumptions; column 3- how it shaped the intervention)- I think this is in the logic model but if in a smaller table it might be easier for the reader to digest.

Is the program theory (theories) outlined anywhere?

The 8 core targeted behaviors- how do these relate to the theories and lit reviews?

Page 7 paragraph 2- "a key findings from both reviews" (what reviews? the lit reviews?)

Page 7 paragraph 3- typos - "agreeing [on] a goal" "agreeing [on] responsibilities"

In the paper the authors note that 6 patient participants were interviewed but the table indicates 9 different participants. In the discussion there is reference to caregivers (did the authors mean family caregivers?) Unclear where and if their views were incorporated.

There is no mention of the ethnic background or socio-economic status of participants. Do the participants characteristics limit the transferability of findings in any way? Should this be described as a limitation?

Why was there such an imbalance between HCPs (n = 20) and patients (n = 6? 9?) Why were family carers not involved, given the significant role that they play in the day to day care of older adults of multimorbidity? Is this a limitation?

How was the qualitative data categorized and by whom? What approach was used?

Why are no quotes included of the body of the paper (I see a table with quotes but the content can be brought to life by pulling some examples into the main text).

In the discussion I would consider incorporating some mention of the need for skills training in prioritization of goals- particularly for providers in dealing with ethical dilemmas and conflict when provider and patient goals/priorities don't align.

6. PLOS authors have the option to publish the peer review history of their article (what does this mean?). If published, this will include your full peer review and any attached files.

Reviewer #1: No

---

## [Author Response · Author response to Decision Letter 0]

8 Jul 2020

Thank you for your very positive and helpful comments. We are very pleased that the paper has been warmly received. We have carefully considered reviewers’ comments, updated the searches and addressed each comment in full as set out below. The page numbers referred to below refer to the page of the unmarked manuscript. 

Review Comments to the Author

Reviewer #1: This is a really interesting and timely paper on the design of an intervention to support person centered care. A number of steps were taken to develop the intervention including two literature reviews, integration of theories and interviews. Due to the number of stages and steps it is a bit confusing how all of the pieces fit together.

- The introduction is clear, compelling and well-written

Thank you for this positive feedback!

- As the paper goes on it becomes very confusing given all of the steps and stages that the authors undertook (I appreciate that this is hard to describe in one paper) but a number of modifications can be made to enhance clarity. I describe below...

We have re-written this to enhance clarity. We have also included a new figure that outlines the development process of the CHAT &PLAN (Figure 1)

- The research questions need to be upfront and clear (not tucked away into a table).

We have moved the aims statement from lines 71-74 to lines 46-48

- All of the figures are blurry and hard to read (these are essential to the paper and need to be fixed)

We have uploaded the images in a different file format for clarity.

-The logic model is unreadable even when I downloaded it as a separate file. It does seem quite text heavy though. Can a more user friendly, less text dense version be prepared? It seems that this is central to the paper and more detail is needed on how it was put together and how the patient partners on the team helped to shape it.

The logic model has been reproduced in a more user friendly, less text dense manner. A statement has been added to clarify: 

This logic model was iteratively designed by the multi-disciplinary study team of co-investigators, with input from our PPI volunteers.

-It appears that a scoping review was conducted as well as an 'in-depth' formal qualitative literature review (submitted for publication)- please describe this method in a few lines. 

We have explained this in greater clarity: “a rapid scoping review of the literature to gather evidence from a broad range of resources about potential intervention features and important contextual factors. This helped us to develop an overview of the topic area and identify key issues that were important to address.”

-For the qualitative lit review, references should be added into that paragraph as well.

The reference for the published review has been added

-In the discussion there is mention of a systematic review (typo?). Was the scoping review published too?

The systematic qualitative review now been referenced here also. The scoping review was not published

-Top of page 6- first sentence has typos."...insufficient of knowledge and skills..." as well as further down "...aim to agree a personalised management plan..."

These sentences have been edited.

-Page 7- paragraph two "a key finding from both reviews"- is a very dense paragraph. The authors then go on to discuss how self-determination theory shaped the intervention. There are two other theories mentioned earlier which are not defined and not linked to the intervention. Perhaps this paragraph can be turned into another table (column 1 - theory; column 2- definition of the theory and its core constructs/assumptions; column 3- how it shaped the intervention)- I think this is in the logic model but if in a smaller table it might be easier for the reader to digest.

We have created Table 2 to help the reader to digest this information 

-Is the program theory (theories) outlined anywhere?

We now clarify that the programme theory is outlined in the logic model

-The 8 core targeted behaviours- how do these relate to the theories and lit reviews?

We have now specified that the ‘CHAT&PLAN’ targeted eight core behaviours based on the guiding principles and theoretical analysis Findings from the reviews are summarised in Table 1, which also demonstrates how they were used to develop intervention guiding principles. Table 1 also links these guiding principles to the aims outlined in the logic model, demonstrating how they informed the intervention development and helped us to identify key context-specific behavioural issues to be addressed.

-Page 7 paragraph 2- "a key findings from both reviews" (what reviews? the lit reviews?)

We have now clarified “both the rapid scoping review of the literature and synthesis of qualitative studies’

-Page 7 paragraph 3- typos - "agreeing [on] a goal" "agreeing [on] responsibilities"

These sentences have been moved to Table 2 and edited.

-In the paper the authors note that 6 patient participants were interviewed but the table indicates 9 different participants.

The “n=6” statement was a typo and we thank the reviewer for highlighting this. This has now been changed to state n=8

-In the discussion there is reference to caregivers (did the authors mean family caregivers?) Unclear where and if their views were incorporated.

We apologise for this lack of clarity. Participants 125 and 130 were family caregivers. We had previously included them in the table marked “Healthcare participants”, as they were also older adults receiving healthcare support. They had not previously had a diagnosis of cancer.

However, for clarity we have now created a separate sub-section of the table for caregivers.

Further, in the discussion we note the following:

Unfortunately, we only recruited 2 caregivers. This is because we aimed to recruit caregivers identified by the older adult participants. The other participants did not identify a caregiver to participate in the study; four of these lived alone.

-There is no mention of the ethnic background or socio-economic status of participants. Do the participants’ characteristics limit the transferability of findings in any way? Should this be described as a limitation?

Yes- in the Strengths and limitations section of the paper we now address this point:

We had initially aimed to recruit more participants in the patient groups. However, despite contacting a number of potentially eligible individuals, many chose not to participate. Difficulties in recruitment may be due to the nature of the conditions we wished to study and the busy health-related workload experienced by those with ill-health[1]. As well as health-status, other factors that could impact recruitment may relate to the ethnicity of participants. All of the participants who chose to participate in this study were white, which may be a reflection of the ethnic diversity in the local area (77.7% of residents recorded their ethnicity as White British in the 2011 Census[2]). However, efforts should be made to gain an insight into the experiences of those of different backgrounds, and to use different methods of approach to encourage participation in research[3]. Moreover, we did not ask specific questions about the socio-economic status of participants, yet it was evident that some were affluent as indicated in conversations about private healthcare and paying for support such as gardeners or cleaners. However, other participants did address financial challenges relating to paying for parking at the hospital and purchasing items such as sanitary pads. 

-Why was there such an imbalance between HCPs (n = 20) and patients (n = 6? 9?) 

In the Strengths and limitations section of the paper we now address this point:

It is worth noting that we did gain an insight into experiences of living with multimorbidity by drawing on the findings of previous research in our qualitative review and synthesis. The findings of the synthesis reflected the views of more than 960 patients and 52 family caregivers. The findings of the review were largely echoed by participants in our study. Further, while it was important to explore the acceptability of the tool in patient groups, our primary focus was on healthcare providers as they were the ones that would use the intervention. These individuals offered a more detailed insight into how the CHAT&PLAN would be implemented into routine practice.

-Why were family carers not involved, given the significant role that they play in the day to day care of older adults of multimorbidity? Is this a limitation?

As noted, we only recruited 2 caregivers. This is because we aimed to recruit caregivers identified by the older adult participants. This may be considered a limitation, however we did gain an insight into the role of caregivers in the synthesis that included studies with input from 52 individual family caregivers. Further, as noted above, in this development study our primary focus was on healthcare providers as they were the ones that would use the intervention in practice.

-How was the qualitative data categorized and by whom? What approach was used?

Thematic analysis was used. This process been outlined in the following paragraph: After interviews were conducted, initial thoughts and ideas were noted down by the interviewers as an early stage of analysis. The data were transcribed verbatim. Data were analysed using thematic analysis to assess participants’ thoughts about the intervention content and inform potential changes. Initial codes were identified and highlighted factors considered pertinent to the design and implementation of the intervention. The generation of initial codes was primarily done by one researcher (TC) with occasional cross-checking to independent coding by a second researcher (AY). Coding was discussed by members of the study team (TC, AY and JB) and developed into themes

-Why are no quotes included of the body of the paper (I see a table with quotes but the content can be brought to life by pulling some examples into the main text).

This was decided to reduce the length of text in the paper and to link the quotes directly to the points that were made. We think this approach aids us in simplifying the presentation of detailed findings from multiple data-sets in one paper but will be able to add in quotes should this be required.

To guide the reader to the quotes, we have edited line 196 to state “Some of the key points to emerge from the interviews are outlined below, and supporting quotes can be found in Table 4.”

-In the discussion I would consider incorporating some mention of the need for skills training in prioritization of goals- particularly for providers in dealing with ethical dilemmas and conflict when provider and patient goals/priorities don't align.

Thank you for this suggestion. We have included a statement in line 385 :

This would include setting and prioritising goals, as well as shared decision-making about goals with service users. This is likely to require more effective communication and coordination between HCPs.

1. Sygna, K., S. Johansen, and C.M. Ruland, Recruitment challenges in clinical research including cancer patients and caregivers. Trials, 2015. 16(1): p. 428.

2. Observatory, S.D. Ethnicity and Language. 2019 09 August 2019 July 2019]; Available from: https://data.southampton.gov.uk/population/ethnicity-language/#:~:text=Resources-,Ethnicity,has%20become%20more%20ethnically%20diverse.

3. Rockliffe, L., et al., It’s hard to reach the “hard-to-reach”: the challenges of recruiting people who do not access preventative healthcare services into interview studies. International journal of qualitative studies on health and well-being, 2018. 13(1): p. 1479582.

---

## [Decision Letter · Decision Letter 1]

29 Sep 2020

Planning and optimising CHAT&PLAN: a conversation-based intervention to promote person-centred care for older people living with multimorbidity

PONE-D-20-07914R1

Dear Dr. Corbett,

We’re pleased to inform you that your manuscript has been judged scientifically suitable for publication and will be formally accepted for publication once it meets all outstanding technical requirements.

Kind regards,

Adam Todd, PhD

Academic Editor

PLOS ONE

Additional Editor Comments (optional):

Thank you for submitting the revisions to the previous paper – the revised version is greatly improved.  The reviewer has made a number of comments about the paper, and possibly how the work should form the basis of two papers.  

The submitted work seeks to describe the planning of a conversation-based intervention, as well as how the intervention was developed and optimised.  In my opinion, the paper achieves this aim, and adds to our understanding in this area; the approach utilised has potential to be used as a template to develop other interventions in this area.  Although I appreciate the paper could be presented as a qualitative paper, and a scoping review paper, the paper uses these as steps to develop and optimise the intervention.  

Reviewers' comments:

Reviewer's Responses to Questions

**Comments to the Author**

1. If the authors have adequately addressed your comments raised in a previous round of review and you feel that this manuscript is now acceptable for publication, you may indicate that here to bypass the “Comments to the Author” section, enter your conflict of interest statement in the “Confidential to Editor” section, and submit your "Accept" recommendation.

Reviewer #1: (No Response)

2. Is the manuscript technically sound, and do the data support the conclusions?

Reviewer #1: No

3. Has the statistical analysis been performed appropriately and rigorously? 

Reviewer #1: N/A

4. Have the authors made all data underlying the findings in their manuscript fully available?

Reviewer #1: No

5. Is the manuscript presented in an intelligible fashion and written in standard English?

Reviewer #1: No

6. Review Comments to the Author

Reviewer #1: Thank you for revising your manuscript. The introduction and discussion read really well. My concern is that there are so many different components and methods in this paper that each one ends up being ill described. I appreciate that the authors want to have everything in one manuscript but it is confusing for the reader and not comprehensive. For example, a whole paper could be dedicated to how the interviews were conducted, analyzed and used to adapt the intervention. As a qualitative researcher who does co-design work, I would love a thorough step by step example of what this looks like. That would be an important contribution. The edits made to the thematic analysis are insufficient. I still don't know how themes were derived, what specific steps were followed, and the names of the themes. Also, the use of terms like "informal scoping review" and "rapid scoping review" and "formal" review left me really confused. No references to traditional methods/ steps were made. It is not clear to me how all of the pieces of the methods fit together. Though the topic is of great importance, there are many holes and as written would not be possible to replicate. I would suggest that this could actually be divided into at least 2 companion papers which take the reader through a more detailed account of how the intervention was adapted and more details in the methods so you can demonstrate the rigour of your work.

7. PLOS authors have the option to publish the peer review history of their article (what does this mean?). If published, this will include your full peer review and any attached files.

Reviewer #1: No

---

## [Editor Report · Acceptance letter]

6 Oct 2020

PONE-D-20-07914R1 

Planning and optimising CHAT&PLAN: a conversation-based intervention to promote person-centred care for older people living with multimorbidity 

Dear Dr. Corbett:

I'm pleased to inform you that your manuscript has been deemed suitable for publication in PLOS ONE. Congratulations! Your manuscript is now with our production department. 

Kind regards, 

on behalf of

Dr. Adam Todd 

Academic Editor

PLOS ONE